# Integration of Repeatomic and Cytogenetic Data on Satellite DNA for the Genome Analysis in the Genus *Salvia* (Lamiaceae)

**DOI:** 10.3390/plants11172244

**Published:** 2022-08-29

**Authors:** Olga V. Muravenko, Olga Yu. Yurkevich, Julia V. Kalnyuk, Tatiana E. Samatadze, Svyatoslav A. Zoshchuk, Alexandra V. Amosova

**Affiliations:** Engelhardt Institute of Molecular Biology, Russian Academy of Sciences, 32 Vavilov St., 119991 Moscow, Russia

**Keywords:** *Salvia*, high throughput sequencing, repeatome, transposable elements, 45S rDNA, 5S rDNA, satellite DNA, FISH chromosome mapping

## Abstract

Within the complicated and controversial taxonomy of cosmopolitan genus *Salvia* L. (Lamiaceae) are valuable species *Salvia officinalis* L. and *Salvia sclarea* L., which are important for the pharmaceutical, ornamental horticulture, food, and perfume industries. Genome organization and chromosome structure of these essential oil species remain insufficiently studied. For the first time, the comparative repeatome analysis of *S. officinalis* and *S. sclarea* was performed using the obtained NGS data, RepeatExplorer/TAREAN pipelines and FISH-based chromosome mapping of the revealed satellite DNA families (satDNAs). In repeatomes of these species, LTR retrotransposons made up the majority of their repetitive DNA. Interspecific variations in genome abundance of Class I and Class II transposable elements, ribosomal DNA, and satellite DNA were revealed. Four (*S. sclarea*) and twelve (*S. officinalis*) putative satDNAs were identified. Based on patterns of chromosomal distribution of 45S rDNA; 5S rDNA and the revealed satDNAs, karyograms of *S. officinalis* and *S. sclarea* were constructed. Promising satDNAs which can be further used as chromosome markers to assess inter- and intraspecific chromosome variability in *Salvia* karyotypes were determined. The specific localization of homologous satDNA and 45S rDNA on chromosomes of the studied *Salvia* species confirmed their common origin, which is consistent with previously reported molecular phylogenetic data.

## 1. Introduction

*Salvia* L. is the largest genus of the family Lamiaceae which includes around 1000 species widely distributed in tropical, subtropical, and temperate regions. The taxonomy of this extensive genus is quite complicated and still remains ambiguous [1,2,3]. This genus is traditionally defined morphologically by an unusual lever-like staminal plant character that is formed with elongated connectives and filaments [4,5]. This staminal morphology is highly diverse, and about 11 stamen types were described within *Salvia* [5,6,7,8]. Further molecular phylogenetic studies generally confirmed the morphological classification of *Salvia* [6,7,8,9,10,11]; and the genus was subdivided into 11 subgenera [3,9,12]. Recently, the analysis of plastid and nuclear genomes using next generation sequencing (NGS) technologies have largely confirmed the polyphyletic origin of this genus and also revealed the monophyletic origin of each of its clades, which strongly correlated with a geographic distribution of *Salvia* species [3,9,13].

Despite the fact that various taxonomic revisions have been presented, the chromosomal evolution of this genus is still poorly understood. Within the genus *Salvia*, basic chromosome numbers were shown to vary significantly (x = 6, 7, 8, 9, 11, 13, 15, 17, and 19), and ploidy levels ranged from diploid to octoploid [14,15,16]. Moreover, in different *Salvia* species, inter-population variability in chromosome numbers was detected [17,18,19]. For a number of *Salvia* species, polyploid populations were revealed along with diploid ones [16,20]. In karyotypes of *S. sclarea*, *S. nemorosa*, *S. farinacea*, *S. reuterana*, and *S. officinalis*, 0–2 supernumerary B chromosomes were detected [21,22,23,24]. The relationship revealed between geographical distribution, chromosome numbers, and ploidy levels indicates that both aneuploidy and polyploidy played an important role in the speciation processes within *Salvia* [16,20].

*Salvia* species have small-sized chromosomes (1–4 μm), which complicates the comprehensive analysis of their morphology. Additionally, most of the cytogenetic studies are based on acetocarmine, aceto-orcein, or hematoxylin staining of chromosomes, and such techniques do not always allow reliable identification of individual chromosomes in karyotypes [16,17,18,20,25,26,27]. In one *Salvia* species, *S. miltiorrhiza*, FISH-based chromosome mapping of 45S and 5S rDNA was performed, and the genome of this species was fully sequenced and annotated by linkage groups [28]. At the same time, the comprehensive molecular cytogenetic characterization of most *Salvia* species is still required to investigate the chromosomal evolution within the genus. It was shown for other plant species that such studies usually include FISH-based chromosome mapping of various molecular markers, e.g., 45S rDNA, 5S rDNA and other tandem repeats (satellite DNA families) [29,30,31,32]. However, the chromosome diversity as well as relationships between genomes within the genus *Salvia* still remain unexplored.

The genus *Salvia* comprises many economically valuable species, including biennial *S. sclarea* L. subg. *Sclarea* sect. *Stenarrhena* subsect. *Gongrosphaceae* and perennial *S. officinalis* L. subg. *Eusalvia* sect. *Eusphace* [1]. Both species are highly aromatic, and they are widely used in the pharmaceutical, ornamental horticulture, food, and perfume industries [33,34,35,36]. Particularly, *S. officinalis* is very important for pharmaceutical purposes. Its inflorescences and leaves contain a large amount (0.3–0.5%) of essential oil, the main components of which are terpenes, flavonoids, and salvin [37,38,39,40]. *S. sclarea* is also rich in essential oil (0.18–0.83%), including such components as linalyl acetate, linalool, germacrene D and sclareol (1.0–1.5%) [34,41]. This plant is widely used in luxury perfumery, winemaking, and medicine [2,34,36]. According to the molecular phylogenetic data, the *S. sclarea* and *S. officinalis* were included in one clade or occupy sister clades within the genus *Salvia* [6,7,8,9,10,11,12,13,42]. At the same time, the analysis of their karyotypes using monochrome staining revealed differences both in the main chromosome numbers and chromosome morphology. It was shown that the karyotype of *S. sclarea* contained 2n = 2x = 22 metacentric and acrocentric chromosomes (0.92–1.67 μm) [43] while in karyotype of *S. officinalis*, 2n = 2x = 14 chromosomes (0.99–2.38 μm) of different morphology were observed [17,26,44]. In order to study the genomics of these *Salvia* species and also identify economically valuable genes, their genomes were explored with the use of high-throughput sequencing methods. Specifically, the screening of BAC library of 17764 clones was performed for *S. officinalis*; the transcriptome of *S. sclarea* was characterized, and SSR markers for economic genes were explored [45,46,47,48]. At the same time, the repeatomes of these species are still poorly investigated.

One of the effective approaches for characterization of repetitive DNA in plant species is a genome-wide bioinformatic analysis by RepeatExplorer/TAREAN (Tandem Repeat Analyzer) pipelines. Such an approach is widely used in comparative repeatome analyses between related species, and it provides new insight into the organization and divergence of their genomes [29,49,50]. Moreover, newly revealed families of satellite DNA (satDNAs) are often used as chromosome markers in comparative molecular cytogenetic studies to assess intra- and interspecific diversity of genomes in various plant taxa [29,30,31,32,50].

In the present study, for the first time, a comparative analysis of repeatomes of two economically valuable species, *S. sclarea* and *S. officinalis*, was carried out based on the obtained NGS data and RepeatExplorer/TAREAN pipelines. Moreover, FISH chromosome mapping of 45S rDNA, 5S rDNA, and the revealed satDNAs was performed to examine their chromosomal distribution, identify new molecular markers and study genome relationships between these species.

## 2. Results

### 2.1. Comparative Analyses of the Repetitive DNA Sequences

The comparative analysis of repeatomes in *S. officinalis* and *S. sclarea* showed that retrotransposons, including Ty3-Gypsy and Ty1-Copia superfamilies (Class I transposable elements), were the most abundant in both species (Table 1; Figure 1). At the same time, the number of the retrotransposons differed greatly between *S. sclarea* and *S. officinalis* (31.33% and 13.2%, respectively). In the repeatome of *S. sclarea*, Ty1-copia (16.28%, SIRE was dominant) were more than twice as abundant as Ty3-gypsy (7.47%, non-chromovirus Athila and chromovirus Tekay were dominant). In *S. officinalis*, however, the Ty3-gypsy retroelements (9.54%, chromovirus Tekay and non-chromovirus Athila were dominant) were roughly six times more abundant than Ty1 Copia retroelements (1.53%, Angela and SIRE were dominant). DNA transposons (Class II) were found in lower amounts compared to retrotransposons. In the repeatome of *S. officinalis*, the quantity of Class II mobile elements was less (0.34%) if compared with *S. sclarea* (0.45%). Additionally, in *S. sclarea*, the genome proportion of unclassified LTR retroelements (7.38%) exceeded those revealed in *S. officinalis* (1.81%).

In repeatomes of the studied species, different proportions of ribosomal DNA (2.07% in *S. sclarea* and 3.01% in *S. officinalis*) and satellite DNA (1.63 % in *S. officinalis* and 3.4% in *S. sclarea*) were revealed (Figure 1, Table 1).

In *S. officinalis*, eight high confident putative satDNAs and four low confident putative satellites were detected. In *S. sclarea*, three high confident putative satDNAs and one low confident putative satDNA were identified. The sequences of the revealed satellite DNAs for both studied species are presented in Appendix A.

### 2.2. BLAST Results

BLAST revealed sequence similarity between SO3, SO97, SO107, and SO108 repeats (69–86.5% of identity), and also between SO3, SO97, SO107, and SS1 repeats (69.4–83.7% of identity) (Appendix A). Additionally, SO4, SO48, SO97, SO108, and SS337 repeats exhibited partial sequence similarity with several DNA repeats identified in genomes of species from other genera including *Ipomoea trifida, Jasminum sambac*, *Theobroma cacao*, and *Arabidopsis arenosa* (detailed in Appendix A).

### 2.3. Chromosomal Structural Variations

The performed karyotype analyses showed that the studied *Salvia* species presented diploid karyotypes with 2n = 2x = 14 (*S. officinalis*) and 2n = 2x = 22 (*S. sclarea*) chromosomes ranged from 2 to 6 µm (Figure 2, Figure 3, Figure 4, Figure 5 and Figure 6). In some karyotypes of *S. officinalis*, 0–2 small additional chromosomes (B chromosomes) were detected (Figure 2A,F and Figure 3C).

In the karyotype of *S. officinalis*, four bright hybridization signals of 45S rDNA were revealed in the short arms of chromosome pairs 2 and 3. In some karyotypes, a cluster of 45S rDNA was observed on one B chromosome (Figure 3C). Four 5S rDNA clusters were observed in the short arms of chromosome pairs 3 and 7 (both in colocalization with 45S rDNA) (Figure 2, Figure 3 and Figure 4 and Figure 7).

In *S. officinalis*, different patterns of chromosome distribution of the oligonucleotide probes were observed (Appendix A, Figure 2, Figure 3, Figure 4 and Figure 7):

(1) SO44 and SO113 presented clustered localization in the pericentromeric region of all the chromosome pairs;

(2) SO3, SO48, SO97, SO107, SO108, SO145, SO202 and SS1 demonstrated clustered localization in the subtelomeric region of the long arm of one or two homologous chromosomes 4. Additionally, small clusters of SO107 were detected in the subtelomeric regions of all chromosome pairs; a minor site of SO202 was revealed in the subtelomeric region of the short arm of one homolog of chromosome pair 6;

(3) Clusters of SO4 were observed in the subtelomeric regions of the long arms of one or two homologous chromosomes 5;

(4) Clusters of SO46 were revealed in the short arm (in co-localization with 45S rDNA and 5S rDNA) of chromosome pair 3; 

(5) SO342 was distributed dispersed along all the chromosome pairs.

In some karyotypes, clusters of SO3, SO145 and SS1 were detected on B chromosomes (Figure 2F, Figure 3C and Figure 7B).

In the karyotype of *S. sclarea*, bright hybridization signal of 45S rDNA was revealed in the short arm of one chromosome pair 2. Clusters of 5S rDNA were localized in the short arms of chromosome pairs 5 and 11 (Figure 5, Figure 6 and Figure 7).

The chromosome distribution patterns of four oligonucleotide probes (SS1, SS8, S123, and SS337) presented clustered localization in the pericentromeric and/or terminal chromosome regions (Appendix A, Figure 5, Figure 6 and Figure 7).

Based on chromosome morphology and distribution patterns of the studied molecular cytogenetic markers, the chromosome pairs in karyotypes were identified, and the species karyograms and schemes demonstrating localization of the examined markers on chromosomes were constructed (Figure 4, Figure 6 and Figure 7).

## 3. Discussion

In the present study, to clarify the genomic relationships between *S. officinalis* and *S. sclarea*, the comparative analysis of their repeatomes was performed for the first time. Plant genomes are known to contain a great number of repetitive DNA sequences [51,52]. They include highly abundant transposable elements (TEs) and tandem repeats, which are diverse genome components [53,54,55]. These elements play an important role in genome organization and evolution, since they can change their location and/or copy numbers [53,56,57]. Based on structural characteristics and mode of replication, TEs are subdivided into two main classes, class I (retrotransposons, including LTR retrotransposons) and class II (DNA transposons) [56,58,59]. LTR retrotransposons are highly abundant, and they can constitute up to 75% of nuclear DNA in plants [60,61]. LTR retrotransposons comprise Ty1-Copia and Ty3-Gypsy superfamilies, which are further subdivided into a large number of families specific to a single or a group of closely related species [59]. In the present study, LTR retrotransposons were also the most abundant elements in repeatomes of both *S. officinalis* and *S. sclarea.* LTR retrotransposons are considered to be the main contributors to nuclear genome changes in angiosperms [62,63,64]. These retroelements are able to replicate generating new copies of themselves through the copy and paste mechanism, thus increasing the size of the genome [54]. At the same time, the LTR copies can also be efficiently eliminated from the genome via both solo LTR formation and accumulation of deletions, which reduces the genome size [60]. Moreover, plants having a small genome size are shown to contain fewer LTR retrotransposons compared to those with large genomes [64,65]. A genome size is considered to be an intrinsic property of a species which, however, varies greatly among plants. Moreover, intra- and interspecific variations in genome size might reflect different evolutionary processes during speciation [64,66]. The average genome size of *S. officinalis* is small (1C = 479.22–572.17 Mbp), and it roughly corresponds to that of *S. sclarea* (1C = 567.24–655.26 Mbp) [67,68,69]. At the same time, between these species, we revealed differences in genome proportions of the total content of retrotransposons, in ratio between Ty3-Gypsy/Ty1-Copia elements, in proportion to other retroelements, including SIRE, Angela, non-chromovirus Tat-Retand, non-chromovirus Athila, chromovirus Tekay, and also the total amount of DNA transposons. In particular, in *S. officinalis*, Ty3-Gypsy retrotransposons were six times more abundant compared to Ty1-Copia elements. In repeatome of *S. sclarea*, however, the number of Ty1-Copia elements was twice as many as Ty3-Gypsy elements. The observed interspecific differences might be related to the processes occurred in their genomes during speciation. As reported earlier, the evolutionary changes occurred in genomes of diploid species of *Melampodium* correlated with variations in the content of the SIRE (Ty1-Copia), Athila (Ty3-Gypsy), and CACTA (DNA transposon) lineages [49]. Our results on the retrotransposon content in the genome of *S. officinalis* were basically consistent with the earlier reported data on repeatomes of *Salvia splendens* and *Salvia miltiorrhiza*. In their repeatomes, Ty3-Gypsy elements were also more abundant (17.59% and 29.83%, respectively) compared to Ty1-Copia (8.70% and 14.77%, respectively) [28,70]. The genomic content of TEs can vary depending on the species, and sometimes even on the genotype (e.g., Asian rice subspecies or maize lines), which could be related to speciation processes and/or environmental adaptation of plants [71,72]. Additionally, in *S. officinalis*, the genome proportion of unclassified LTR retroelements exceeded that revealed in *S. sclarea*. These genome differences could be related to the specific attributes within *Salvia*, which highlights the need for more research on these TEs.

Tandem repeats, including rDNA and other satDNAs, are a fast-evolving fraction of plant repeatomes. Even between closely related species, they can vary in copy number and nucleotide composition, as well as in their abundance and distribution in the genome [66]. Between the studied species, interspecific variations in genome proportion of ribosomal DNA were also detected. Nucleolar organizer regions (NORs) of chromosomes are known to contain tandemly repeated 45S rDNA sequences [73]. The revealed variations in genome proportion of ribosomal DNA could be related to the different number of satellite chromosomes bearing NORs in karyotypes of *S. sclarea* (one pair) and *S. officinalis* (two pairs). Additionally, variations in total contents of satellite DNA were revealed between *S. sclarea* (3.4%) and *S. officinalis* (1.63). It is known that satDNA have a variable length of the repeating unit (monomer) and usually form tandem arrays up to 100 Mb [74,75]. Although they are considered to be non-coding sequences, satellite monomers are generally 160–180 bp or 320–370 bp in length (though other lengths can also be found in plants), which corresponds to the length of mono- and dinucleosomes [76,77,78]. The sequences of satellite monomers evolve concertedly through the process of molecular drive; mutations are homogenized in a genome and become fixed in the populations [79]. The sequence identity within an array develops in accordance with a process called “concerted evolution”, which leads to the maintenance of homogeneity of satDNA monomers within a species during evolution [80]. The abundance of satDNA can vary in plant genomes even between generations resulting in high polymorphism in the length of satellite arrays [79]. At the same time, some satDNA sequences demonstrate sequence conservation over long evolutionary periods [81]. Since there are many satellite DNAs in the genome, the evolution of species-specific satDNA may be the result of a change in the number of copies in the library of satellite sequences common to a group of species [78,79,81]. High-throughput DNA sequencing and subsequent genome-wide bioinformatic analysis provide important data on the structural diversity of satDNA [81,82,83]. In the present study, in *S. officinalis*, 12 putative satDNAs were revealed by genomic analyses with TAREAN while only four putative satDNAs were detected in the repeatome of *S. sclarea*. These interspecific differences indicated a high level of satDNA diversity in *Salvia* genomes.

Despite the fact that satDNAs are considered to be rapidly evolving fractions of a genome, some of the fractions have highly conserved monomeric sequences and can persist over long evolutionary periods. This may be due to their interactions with specific proteins required for the formation of heterochromatin, as well as their putative regulatory role in gene expression [79,84]. It is known that satDNAs contribute to the main processes of formation of most important chromosomal structures, such as DNA packaging and chromatin condensation [78,85,86]. In the present study, BLAST detected regions of local sequence similarity between repeats SO3, SO97, SO107, SO108 and SS1, suggesting the common ancient origin of *S. officinalis* and *S. sclarea*, which is consistent with previous molecular phylogenetic data [3,9,13].

Since satDNAs are often associated with heterochromatin regions, they are localized in the certain chromosome regions (centromeric, terminal, and/or intercalary). The cytogenomic analysis of satDNAs is important for understanding of molecular mechanisms of chromosome rearrangements, changes of basic chromosome numbers, and karyotype diversification during genomes evolution of plant species [87,88]. During the processes of speciation and karyotype evolution, basic chromosome number is one of the most important parameters. In the genus *Salvia*, various chromosome numbers were detected (x = 6, 7, 8, 9, 11; 13; 15; 17; and 19) but the most common basic chromosome numbers were x = 7, x = 8, and x = 11. [23,26,43]. Currently, the ancestral basic number is considered to be x = 7 [16,89]. In this regard, the process of karyotype evolution appeared to be from x = 7 to x = 15 and x = 16. The basic number of x = 8 was probably shaped from x = 7 through aneuploidy or dysploidy were also observed [89]. The karyotype evolution of the other basic numbers such as x = 9, x = 10, and x = 11 might occur by descending or ascending dysploidy and dibasic polyploidy [16]. The basic number x = 6 could be involved in the amphipolyploid origin of the basic number x = 11 [90]. The basic numbers of x = 13 and x = 15 might be results of hybridization processes between parents having different chromosome numbers, e.g., the basic number, x = 13 could be a dibasic one shaped from the combination of x = 6 and x = 7 [21]. In the present study, the most common basic chromosome numbers x = 7 in *S. officinalis* was observed while in *S. sclarea*, x = 11 was revealed, which was in accordance with the previous data on the chromosome number evolution in the genus *Salvia*. Moreover, we detected 0–2 B chromosomes in karyotypes of *S. officinalis*. These supernumerary chromosomes were revealed earlier in several *Salvia* species including *S. sclarea* and *S. officinalis* [21,22,23,24]. B chromosomes are found in many eukaryotic organisms. They do not pair with any of the chromosomes of the basic set (A chromosomes), and the distribution of these chromosomes within an individual or among different individuals of a population is not uniform [91,92,93]. The morphology of B chromosomes differs from that of A chromosomes, and they vary in structure and chromatin properties [92]. Considerable structural polymorphism of B chromosomes was reported in a number of plants, e.g., *Aegilops speltoides* [94] and *Brachycome dichromosomatica* [95]. Due to their dispensable nature, B chromosomes are generally considered to be non-essential and genetically inert elements [91]. However, some phenotypic and functional effects of these additional elements were also reported, which are usually dependent upon their numbers. For example, the increased number of B chromosomes in maize mostly has a negative effect on the fertility of organisms [91]. Additionally, B chromosomes were shown to influence the A-genome transcription in maize with the stronger effect associated with an increase in their copy number [96].

Chromosome markers developed based on tandem repeats are particularly useful for identification of chromosomes in karyotypes and also for the analysis of chromosome rearrangements as well as evolution of plant genomes [29,30,31,32,97]. The comparison of patterns of chromosomal distribution of 45S rDNA, 5S rDNA and the revealed satDNAs allowed us to identify all chromosome pairs in karyotypes of both studied species and construct their chromosome karyograms. The identification of all homologous chromosome pairs in karyotype of *S. officinalis* can be performed by multicolor FISH using a set of markers: 45S rDNA, 5S rDNA, SO4, SO113, and SO202. In *S. sclarea*, however, tandem repeats SS1, SS8, SS123 and SS337 were localized in pericentromeric and/or terminal regions of all chromosome pairs. Nevertheless, based on chromosome morphology and chromosome distribution patterns of 45S rDNA, 5S rDNA, SS1, SS8, SS123, and SS337, we were able to construct *S. sclarea* karyograms. However, for more precise chromosome identification in karyotype of *S. sclarea*, further search for molecular chromosome markers including microsatellites, should be carried out [45,48].

As described previously, satDNAs could demonstrate various patterns of distribution of plant chromosomes including their clustered, dispersed and/or combined localization [81,98,99,100,101]. In *S. officinalis*, we observed large clusters of SO3, SO48, SO97, SO107, SO108, SO145, and SO202 predominantly localized in the same terminal region of the long arm of chromosome pair 4. Interestingly, SO3, SO97, SO107and SO108 demonstrated sequence similarity while BLAST homology among SO48, SO145 and SO202 was not revealed. Localization of seven different satDNA in the one and the same chromosome region is not typical for plants. At the same time, each SO4 and SO46 satDNAs presented one cluster in the terminal region of the short arms of chromosome pair 5 and 2, respectively, and such chromosome localization of satDNAs was earlier described for other plant species [102]. Other repeats demonstrated pericentromeric (SO44 and SO113) or dispersed (SO342) chromosome patterns, which is typical for plants [98,99,100,103,104]. In karyotype *S. sclarea*, similar localization of four non-homologous tandem repeats (SS1, SS8, SS123 and SS337) was revealed in pericentromeric and/or terminal regions. Especially interestingly, the SS1 repeat, having predominantly terminal localization on all chromosomes of *S. sclarea* and demonstrating a high level of homology with SO3, SO97, and SO107, was localized in the same regions of chromosome pair 4 and one B chromosome (in co-localization with SO3, SO145 and SO202). The revealed peculiarities of chromosome distribution of SS1 in both studied species showed that reorganization of their chromosome numbers could be related to active chromosomal rearrangements.

The patterns of chromosomal distribution of satDNAs facilitate the recognition of homologous chromosome pairs and recombination as well as differences between lineages and species [74,85]. SatDNA repeats were reported to be recombination “hotspots” of genome reorganization. The localization of satDNAs in the interstitial and/or telomeric heterochromatin regions reduces genetic recombination in the adjacent chromosome regions [105]. Using molecular cytogenetic methods, it was shown that 5S rDNA, 45S rDNA and different satDNAs can be localized both on A and B chromosomes [32,95,101,106]. Further characterization of B chromosomes by NGS technologies revealed that they can include different sequences derived from A chromosomes, 5S rDNA, 45S rDNA, transposons, transcribed genes and various types of repetitive elements [91,93,95]. In this study, we detected 45S rDNA clusters on A and B chromosomes. Moreover, we observed the co-localization of several satDNAs on heteromorphic chromosome pair 4 and also on another B chromosome. These features indicate that the formation of B chromosomes in the *S. officinalis* genome might be related to the changes occurred in the chromosomes of the basic set during speciation.

The structure of the 45S rDNA family is considered to be evolutionary conservative in many taxa, which makes it possible to use 45S rDNA as a standard chromosome marker in genomic relationship studies [107,108]. Notably, the chromosome pair 2 bearing a 45S rDNA cluster had the similar morphology in both studied species. This feature, and also peculiar localization of homologous repeats on chromosomes of the studied *Salvia* species also confirmed the common origin of these species in agreement with previous molecular phylogenetic data [6,7,8,9,10,11,12,13]. Our findings show that satellite DNA plays an important role in the processes of evolution and chromosome diversification in *Salvia* species, and this is essential for elucidation of their origin. At the same time, further comprehensive genomic studies of inter- and intraspecific variability of satDNA arrays are required to explore functional and structural features of other *Salvia* genomes.

## 4. Materials and Methods

### 4.1. Plant Material

Seeds of *S. officinalis* (189375, Bordeaux, France, 2007) and *S. sclarea* (177142, Trieste, Italy, 2000) were obtained from the collections of the Botanic Gardens of Bordeaux, France and Trieste, Italy, respectively. The seeds of the studied accessions were germinated in Petri dishes on the moist filter paper for 3–5 days at room temperature (RT). Then the plants were grown in a greenhouse at 15 °C.

### 4.2. Genomic DNA Extraction and Sequencing

Genomic DNA of *S. officinalis* and *S. sclarea* were isolated from young leaves using the GeneJet Plant Genomic DNA Purification Kit (Thermo Fisher Scientific, Vilnius, Lithuania). The quality of the DNA samples was checked with the Implen Nano Photometer N50 (Implen, Munich, Germany). The concentration and purification of the extracted DNAs were assessed with the Qubit 4.0 fluorometer and Qubit 1X dsDNA HS Assay Kit (Termo Fisher Scientific, Eugene, OR, USA).

For *S. officinalis* and *S. sclarea*, genome DNA low-coverage sequencing was performed at the Beijing Genomics Institute (BGISeq platform) (Shenzhen, Guangdong, China) according to the NGS protocol for generating 5 million of paired-end reads of 150 bp in length, which was at least 1.1–1.5x of the coverage of these *Salvia* genomes [67,68,69]. The raw sequencing data for *S. officinalis* and *S. sclarea* are available in the NCBI (National Center for Biotechnology Information) BioProject database under accession number PRJNA861221 (https://submit.ncbi.nlm.nih.gov/subs/bioproject/SUB11837398/overview).

### 4.3. Sequence Analysis and Identification of DNA Repeats

The genome sequences of *S. officinalis* and *S. sclarea* were used for genome-wide comparative analyses. Interspecific comparisons, reconstruction, and quantification of major repeat families were performed with the use of RepeatExplorer 2 and TAREAN pipelines [109,110]. For each studied species, the genomic reads were filtered by quality, and then 1,000,000 high quality reads were randomly selected for further analyses, which corresponds to 0.23–0.3x of a coverage of the *Salvia* genomes: *S. officinalis* having 1C = 479.22–572.17 Mbp [67,69] and *S. sclarea* with 1C = 567.24–655.26 Mbp [68,69]. That was within the limits recommended by the developers of these programs (genome coverage of 0.01–0.50x was recommended) [110]. RepeatExplorer/TAREAN was launched with the preset settings based on Galaxy platform (https://repeatexplorer-elixir.cerit-sc.cz/galaxy/). The default threshold was explicitly set to 90% sequence similarity spanning at least 55% of the read length (in the case of reads differing in length it was applied to the longer one). The sequences of most abundant satDNAs revealed in the repeatomes of *S. officinalis* and *S. sclarea* are presented in Appendix A. The sequence homology of the revealed satDNAs was estimated by BLAST (NCBI, Bethesda, MD, USA) (Appendix A). Based on the twelve satDNAs of *S. officinalis* (SO) and four satDNAs of *S. sclarea* (SS), oligonucleotide FISH probes SO3, SO4, SO44, SO46, SO48, SO97, SO107, SO108, SO113, SO145, SO202, and SO342, and also SS1, SS8, SS123, and SS337 were generated by the Primer3-Plus software (Appendix A) [111].

### 4.4. Chromosome Spread Preparation

Chromosome preparations were obtained according to the method developed previously for plants with small chromosomes [112]. Plant roots (0.5–1 cm long) were excised and placed in ice-cold water with 1 µg/mL of DNA intercalator 9-AMA (9-aminoacridine) (Sigma, St. Louis, MO, USA) for 24 h to obtain elongated chromosomes. Then, the roots were fixed in the ethanol and glacial acetic acid fixative (3:1) for 2 days at room temperature. The fixed roots were placed in the 1% acetocarmine solution (in 45% acetic acid) for 20 min. Each root was transferred to a glass slide; the root meristem was cut from the tip cap and macerated in a drop of 45% acetic acid. Then, a squashed preparation was made with the use of a cover slip. After freezing in liquid nitrogen, the cover slip was removed; and the obtained preparation was dehydrated in 96% ethanol for 3 min and air dried for 15 min.

### 4.5. Fluorescence In Situ Hybridization

In FISH assays, we used two wheat DNA probes: pTa71 enclosing 18S-5.8S-26S (45S) rDNA [113] and pTa794 containing 5S rDNA [114]. These DNA probes were labelled directly with fluorochromes Aqua 431 dUTP, Red 580 dUTP or Green 496 dUTP (ENZO Life Sciences, NY, USA) by nick translation according to manufacturers’ protocols. Also, oligonucleotide probes SO1, SO3, SO4, SO44, SO46, SO48, SO67, SO97, SO107, SO108, SO113, SO145, SO202, SO342, SS1, SS8, SS123, SS165, SS181, and SS337 were used (Appendix A). These probes were produced and labelled directly with 6-FAM- or Cy3-dUTP in *Syntol* (Moscow, Russia).

Several sequential FISH procedures were performed with various combinations of these labelled DNA probes as described previously [115,116]. Before the first FISH procedure, chromosome slides were pretreated with 1 mg/mL RNase A (Roche Diagnostics, Mannheim, Germany) in 2xSSC at 37 °C for 1 h. Then, the slides were washed three times for 10 min in 2x SSC, dehydrated through a graded ethanol series (70%, 85% and 96%) for 3 min each and air dried for 15 min. Then, 15 µL of hybridization mixture containing 40 ng of each labeled probe was added to each slide, and the slide was covered with a coverslip, sealed with rubber cement, denatured at 74 °C for 5 min, chilled on ice and placed in a moisture chamber at 37 °C. After overnight hybridization, the slides were washed in 0.1xSSC (10 min, 44 °C), twice in 2xSSC for10 min at 44 °C, followed by a 5-min wash in 2xSSC and three 3-min washes in PBS at room temperature. Then, the slides were dehydrated through the graded ethanol series for 3 min each, air dried for 15 min and stained with DAPI (4’,6-diamidino-2-phenylindole) dissolved (0.1 μg/mL) in Vectashield mounting medium (Vector Laboratories, Burlingame, CA, USA). After documenting FISH results, the chromosome slides were washed twice in 2xSSC for 10 min. Then, sequential FISH procedures were conducted on the same slides.

### 4.6. Chromosome Analysis

The chromosome slides were inspected using the epifluorescence Olympus BX61 microscope with the standard narrow band pass filter set and UPlanSApo 100x/1.40 oil UIS2 objective (Olympus, Tokyo, Japan). Chromosome images were captured with a monochrome CCD (charge-coupled device) camera (Snap, Roper Scientific, Tucson, AZ, USA) in grayscale channels, pseudo-coloured and processed with Adobe Photoshop 10.0 (Adobe Systems, Birmingham, AL, USA) and VideoTesT-FISH 2.1 (IstaVideoTesT, St. Petersburg, Russia) software. At least five plants and 15 metaphase plates were examined in each sample. Chromosome pairs in karyotypes were identified according to the chromosome size and morphology and localization of the chromosome markers.

## Figures and Tables

**Figure 1 plants-11-02244-f001:**
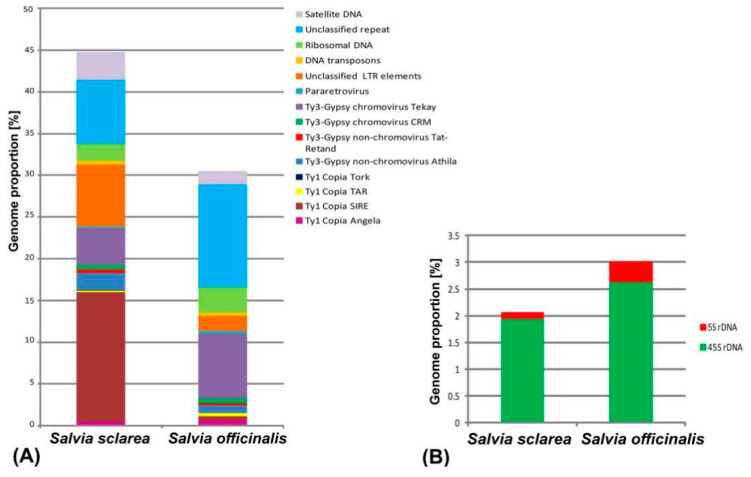
Proportions of the most abundant repetitive DNA sequences (**A**) and 5S and 45S rDNA (**B**) identified in genomes of *Salvia officinalis and Salvia sclarea*. Each genome proportion of the individual repeat type was calculated as a ratio of reads specific to individual repeat types to all reads used for clustering analyses by the RepeatExplorer pipeline.

**Figure 2 plants-11-02244-f002:**
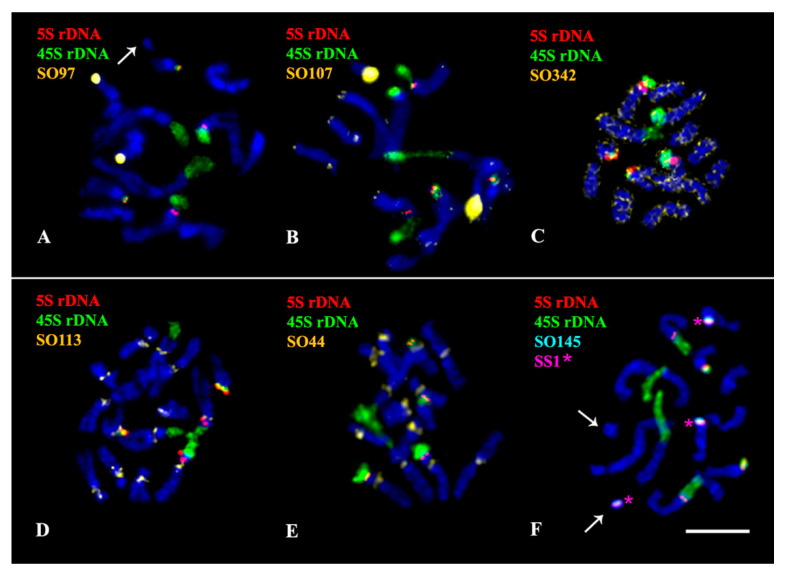
Localization of 45S rDNA, 5S rDNA, and oligonucleotide-based probes SO97 (**A**), SO107 (**B**), SO342 (**C**), SO113 (**D**), SO44 (**E**), and also SO145 and SS1 (**F**) on chromosomes of *Salvia officinalis*. Merged fluorescent images after multicolor FISH with different combinations of the probes. Probe colors are shown on the left. Chromosome DAPI-staining—blue. Asterisks indicate clusters of SS1 colocalized with SO145. Arrows point to B chromosomes. Scale bar—5 µm.

**Figure 3 plants-11-02244-f003:**
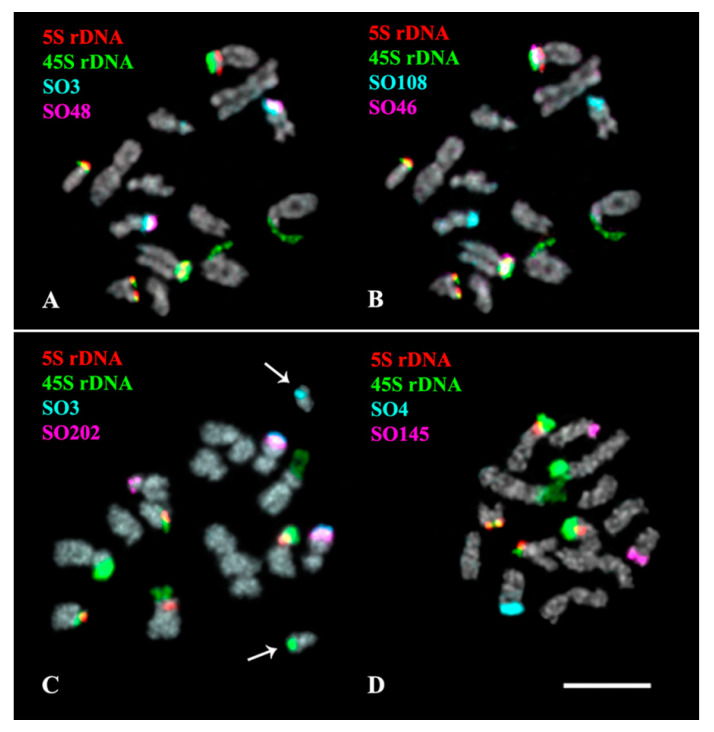
Localization of 45S rDNA, 5S rDNA, and oligonucleotide-based probes SO3 and SO48 (**A**), SO108 and SO46 (**B**), SO202 and SO3 (**C**), and also SO4 and SO145 (**D**) on chromosomes of *Salvia officinalis*. Merged fluorescent images after multicolor FISH with different combinations of the probes. Probe colors are shown on the left. Chromosome DAPI-staining—grey. Arrows indicate B chromosomes. Scale bar—5 µm.

**Figure 4 plants-11-02244-f004:**
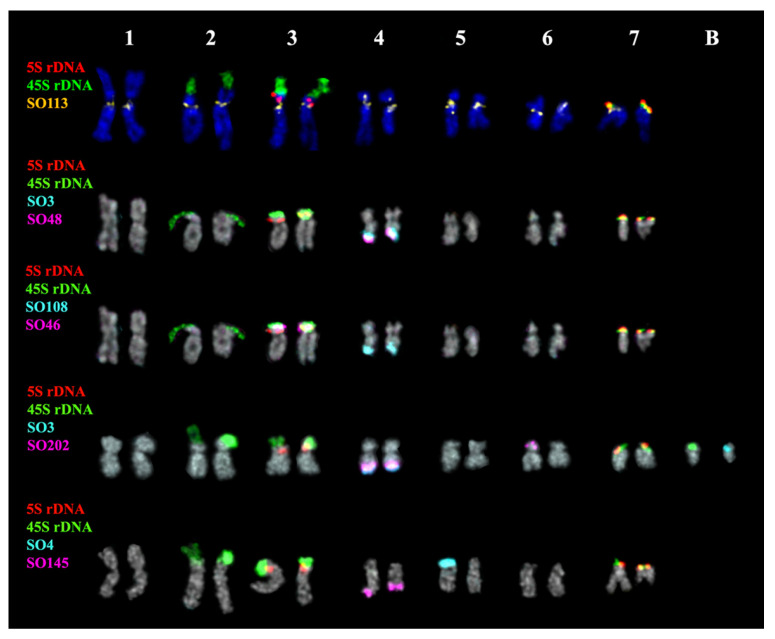
Karyograms of *Salvia officinalis* after multicolor FISH with 45S rDNA (green), 5S rDNA (red), SO113 (yellow), SO3, SO4, and SO108 (aqua), and also SO48, SO46, SO202, and SO145 (purple) (the same metaphase plates as in Figure 2D and Figure 3A–D). Chromosome pair numbers—1–7. B–B chromosomes.

**Figure 5 plants-11-02244-f005:**
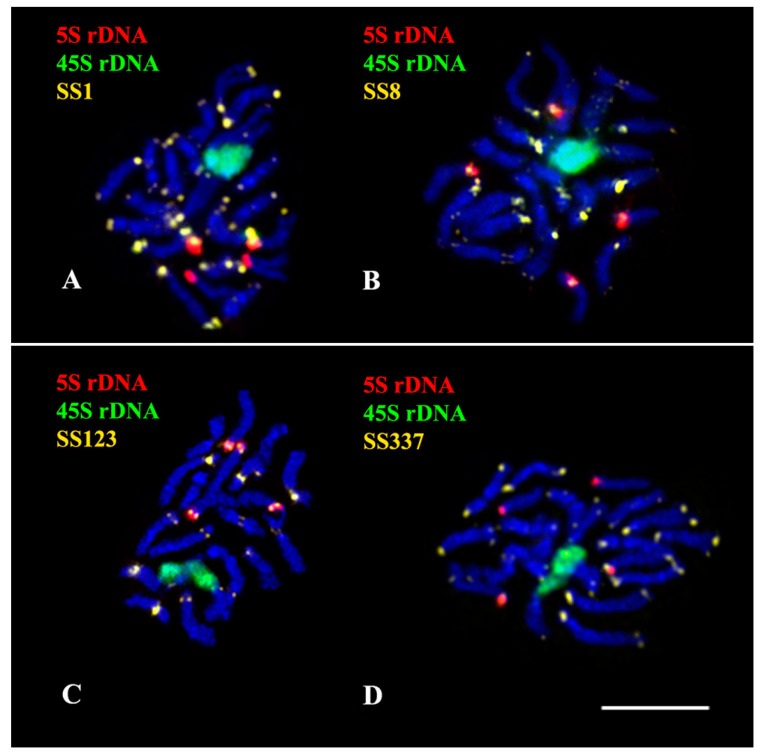
Localization of 45S rDNA, 5S rDNA, and oligonucleotide-based probes SS1 (**A**), SS8 (**B**), SS123 (**C**), and SS337 (**D**) on chromosomes of *Salvia sclarea*. Merged fluorescent images after multicolor FISH with different combinations of the probes. Probe colors are shown on the left. Chromosome DAPI-staining—blue. Scale bar—5 µm.

**Figure 6 plants-11-02244-f006:**
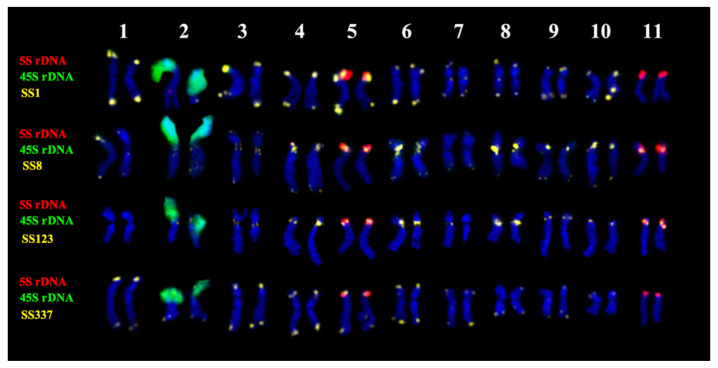
Karyograms of *Salvia sclarea* after multicolor FISH with 45S rDNA (green), 5S rDNA (red), and also SS1, SS8, SS123 and SS337 (yellow) (the same metaphase plates as in Figure 5A–D). Chromosome pair numbers—1–11.

**Figure 7 plants-11-02244-f007:**
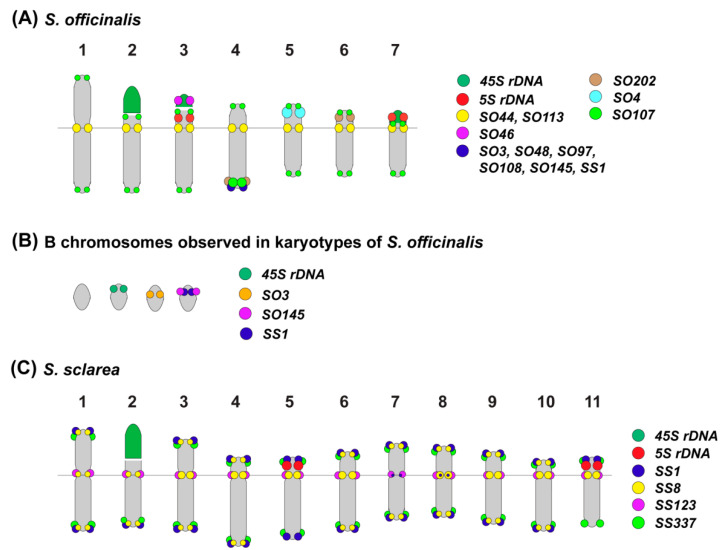
Chromosome schematic representation of the *Salvia* species karyotypes and revealed B chromosomes (**A**–**C**), showing the positions of the oligonucleotide-based probes. Probe colors are shown on the right.

**Table 1 plants-11-02244-t001:** Proportion of major repetitive DNA sequences identified in genomes of *S. officinalis* and *S. sclarea*.

Repeat Name	Genome Proportion (%)	
	*Salvia sclarea*	*Salvia officinalis*
**Retrotransposons** (**Class I**)	**31.33**	**13.2**
**Ty1-Copia**	**16.28**	**1.53**
Angela	0.48	0.62
Bianca	0.03	-
Ikeros	0.03	0.02
SIRE	15,49	0.51
TAR	0.22	0.36
Tork	0.03	0.02
**Ty3-Gypsy**	**7.47**	**9.54**
Non-chromovirus Athila	2.05	0.92
Non-chromovirus Tat-Retand	0.4	0.22
Chromovirus CRM	0.61	0.61
Chromovirus Tekay	4.41	7.79
**Pararetrovirus**	**0.2**	**0.32**
**Unclassified LTR elements**	**7.38**	**1.81**
**Transposons** (**Class II**)	**0.45**	**0.34**
EnSpm_CACTA MuDR_Mutator	0.25 -	0.3 0.04
PIF_Harbinger	0.15	-
Helitron	0.05	-
**Ribosomal DNA**	**2.07**	**3.01**
45S rDNA	1.94	2.62
5S rDNA	0.13	0.39
**Unclassified repeats**	**7.62**	**12.6**
**Satellite DNA**	**3.4**	**1.63**
**Organelle**	**13.78**	**15.25**
**Putative satellites**	**3 high confident** **1 low confident**	**8 high confident** **4 low confident**

## Data Availability

The data presented in this study are contained within the article and Appendix A. The datasets of raw sequencing data generated during this study for *S. officinalis* (SAMN29888211) and *S. sclarea* (SAMN29888212) are available in the NCBI (National Center for Biotechnology Information) BioProject database under accession number PRJNA861221 (https://submit.ncbi.nlm.nih.gov/subs/bioproject/SUB11837398/overview).

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
