# Peer review of "Integration of Repeatomic and Cytogenetic Data on Satellite DNA for the Genome Analysis in the Genus *Salvia* (Lamiaceae)"

_plants, 2022, doi:10.3390/plants11172244_

Round 1

Reviewer 1 Report

Dear Respected Editor,

The manuscript “Integration of Repeatomic and Cytogenetic Data on Salellite DNA for the Genome Analysis in the Genus Salvia (Lamiaceae)” aimed to compare the repeat sequences of two species, Salvia sclarea and Salvia officinalis based on NGS and RepeatExplorer/TAREAN pipelines. The topic is interesting, and the subject matter is of interest to the journal readership, however, there are main problems in the manuscript. The first issue is the language of writing. The English language at many places needs extensive editing to make the manuscript readable and understandable. Second, I noticed that the manuscript didn’t include any statistical analyses. How you determined the significance of your results and assessed the differences between these two species?

Please correct the word “salellite” in the title to “satellite”

Author Response

Dear Editorial Board Team,

Thank you for the email dated 08 August 2022 enclosing the decision and the reviewer’s comments on Manuscript plants-1865094 entitled "Integration of Repeatomic and Cytogenetic Data on Satellite DNA for the Genome Analysis in the Genus Salvia (Lamiaceae)”.

We acknowledge the reviewers for their constructive comments and suggestions very much. We have revised the manuscript accordingly, and our responses are given in a point-by-point manner below. We have also checked English language throughout the manuscript (shown in yellow), and we hope that now it meets the standards of ‘Plants’. Our manuscript with tracked changes has been applied. The changes to the manuscript are shown in yellow and green.

Sincerely yours,

Alexandra V. Amosova

Response to the Reviewer's Comments and Suggestions

Reviewer 1

Dear Respected Editor,

The manuscript “Integration of Repeatomic and Cytogenetic Data on Salellite DNA for the Genome Analysis in the Genus Salvia (Lamiaceae)” aimed to compare the repeat sequences of two species, Salvia sclarea and Salvia officinalis based on NGS and RepeatExplorer/TAREAN pipelines. The topic is interesting, and the subject matter is of interest to the journal readership, however, there are main problems in the manuscript. The first issue is the language of writing. The English language at many places needs extensive editing to make the manuscript readable and understandable. Second, I noticed that the manuscript didn’t include any statistical analyses. How you determined the significance of your results and assessed the differences between these two species?

Please correct the word “salellite” in the title to “satellite”

Answer: We would like to thank the reviewer for valuable comments and thoughtful suggestions.

  • We checked English language throughout the manuscript (shown in yellow).
  • In the title, the word “salellite” has been replaced to “satellite”.
  • We believe that the statistical data analysis is not required for this study. The comparative integrated bioinformatic analysis of repeatomes of Salvia sclarea and Salvia officinalis was performed using RepeatExplorer/TAREAN pipelines according to the protocol of the developers of this software [Novak et al., 2013; 2017]. Besides, sequence homology between the DNA repeats revealed in repeatomes of sclarea and S. officinalis was assessed by BLAST (standard software which is also not required statistical data analysis). Such an approach is widely used in comparative repeatome analyses between related plant species as it provides new insight into the organization and divergence of their genomes [Liu et al., 2019 doi: 10.1186/s12870-019-1769-z; Zwyrtková, J., Němečková, A. et al., 2020 doi:10.1186/s12870-020-02495-0; McCann, et al., 2020. doi: 10.3389/fpls.2020.00362].

To clarify this, we added a small paragraph to the sect. Introduction. Lines 86-93.

Moreover, to assess the interspecific variations and/or relationships between genomes of the studied species, we performed comparative karyotype analysis which included FISH chromosome mapping of standard (45S rDNA, 5S rDNA) and novel (revealed satDNAs) chromosome markers. The constructed (for the first time) karyograms of S. sclarea and S. officinalis are the illustrative demonstration of the differences/common features between these two species. This approach is also often used for evaluating intra- and interspecific genome diversity in various plant taxa [Pamponét et al., 2019, doi: 10.1186/s12864-019-5576-6; Ávila Robledillo et al., 2020, doi: 10.1093/molbev/msaa090; Waminal et al., 2021, doi:10.3389/fpls.2021.629898; Campomayor et al., 2021, doi:10.3390/cells10092358].

Reviewer 2

 Comments and Suggestions for Authors

The authors analyzed the “ Integration of Repeatomic and Cytogenetic Data on Salellite 2 DNA for the Genome Analysis in the Genus Salvia (Lamiaceae)” using proper experimental procedures, presented the results clearly and karyotyped two economically important species, Salvia officinalis L. and Salvia sclarea L.. The manuscript it is very well written and fluid and will contribute for a deep knowledge of the chromosomal structure and genome organization of these plant species. Nonetheless, some minor corrections/ additions of text, listed below, can be made to improve the quality of this manuscript:

1 – Please correct the word ‘satellite’ in the title;

Answer: We would like to thank the reviewer for helpful comments on our manuscript. In the title, the word “salellite” has been replaced to “satellite”.

2 - In Lines 406-407: the authors referred that performed a treatment of the root tips in ice cold water containing “1 lg/mL of 9-AMA” for 16-24h. Given the reduced length of the roots please confirm if the entire root instead of the ‘tip’ (root apex) was treated in ice cold water with 9-AMA, besides, in line 411 the authors referred that ‘the root meristem was cut’; please verify the units of concentration ‘lg/mL’; please indicate the 9-AMA full name. The time period of roots treatment is long (16 to 24 h), please explain why or indicate the required duration specific to each species or both in order to enable the repetition of the experiment and achievement of similar results.

Answer: This part of the text has been modified. Lines 409-415.

3 – In Line 429: between the air drying of the chromosome spreads and probes hybridization it is not missing the chromosomal denaturing step? Particularly for FISH procedures performed with the 45S and 5S rDNA probes, given that it was unnecessary for hybridization with oligonucleotide probes.

Answer: FISH procedures were performed as described in the section Material and Methods. All the DNA probes used in the experiment (including 45S rDNA and 5S rDNA ones) were denatured together on one glass slide (covered with a coverslip and sealed with rubber cement) at 74 °C for 5 min. Then, this slide was chilled on ice and placed in a moisture chamber at 37 °C overnight.

4 – The hybridization conditions for the oligonucleotide probes were the same as for the rDNA probes?

Answer: The FISH conditions were the same. These conditions are suitable for successful hybridization of oligonucleotide probes (having short sequences) because they are more sensitive to denaturation temperature. Under these conditions, rDNA probes readily hybridize on plant chromosomes.

Reviewer 2 Report

The authors analyzed the “ Integration of Repeatomic and Cytogenetic Data on Salellite 2 DNA for the Genome Analysis in the Genus Salvia (Lamiaceae)” using proper experimental procedures, presented the results clearly and karyotyped two economically important species, Salvia officinalis L. and Salvia sclarea L.. The manuscript it is very well written and fluid and will contribute for a deep knowledge of the chromosomal structure and genome organization of these plant species. Nonetheless, some minor corrections/ additions of text, listed below, can be made to improve the quality of this manuscript:

1 – Please correct the word ‘satellite’ in the title;

2 - In Lines 406-407: the authors referred that performed a treatment of the root tips in ice cold water containing “1 lg/mL of 9-AMA” for 16-24h. Given the reduced length of the roots please confirm if the entire root instead of the ‘tip’ (root apex) was treated in ice cold water with 9-AMA, besides, in line 411 the authors referred that ‘the root meristem was cut’; please verify the units of concentration ‘lg/mL’; please indicate the 9-AMA full name. The time period of roots treatment is long (16 to 24 h), please explain why or indicate the required duration specific to each species or both in order to enable the repetition of the experiment and achievement of similar results.

3 – In Line 429: between the air drying of the chromosome spreads and probes hybridization it is not missing the chromosomal denaturing step? Particularly for FISH procedures performed with the 45S and 5S rDNA probes, given that it was unnecessary for hybridization with oligonucleotide probes.

4 – The hybridization conditions for the oligonucleotide probes were the same as for the rDNA probes?

Author Response

(The authors gave the same response as above.)
